# Comprehensive Evidence of Carrier-Mediated Distribution of Amantadine to the Retina across the Blood–Retinal Barrier in Rats

**DOI:** 10.3390/pharmaceutics13091339

**Published:** 2021-08-26

**Authors:** Yusuke Shinozaki, Shin-ichi Akanuma, Yuika Mori, Yoshiyuki Kubo, Ken-ichi Hosoya

**Affiliations:** Department of Pharmaceutics, Graduate School of Medicine and Pharmaceutical Sciences, University of Toyama, 2630 Sugitani, Toyama 930-0194, Japan; m2061220@ems.u-toyama.ac.jp (Y.S.); akanumas@pha.u-toyama.ac.jp (S.-i.A.); s1860308@ems.u-toyama.ac.jp (Y.M.); kubo.yoshiyuki.jf@teikyo-u.ac.jp (Y.K.)

**Keywords:** amantadine, blood–retinal barrier, drug delivery, retinal disease, NMDA receptor, inner BRB, retinal capitally endothelial cells, outer BRB, retinal pigment epithelial cells, transporter

## Abstract

Amantadine, a drug used for the blockage of NMDA receptors, is well-known to exhibit neuroprotective effects. Accordingly, assessment of amantadine transport at retinal barriers could result in the application of amantadine for retinal diseases such as glaucoma. The objective of this study was to elucidate the retinal distribution of amantadine across the inner and outer blood–retinal barrier (BRB). In vivo blood-to-retina [^3^H]amantadine transport was investigated by using the rat retinal uptake index method, which was significantly reduced by unlabeled amantadine. This result indicated the involvement of carrier-mediated processes in the retinal distribution of amantadine. In addition, in vitro model cells of the inner and outer BRB (TR-iBRB2 and RPE-J cells) exhibited saturable kinetics (*K*_m_ in TR-iBRB2 cells, 79.4 µM; *K*_m_ in RPE-J cells, 90.5 and 9830 µM). The inhibition of [^3^H]amantadine uptake by cationic drugs/compounds indicated a minor contribution of transport systems that accept cationic drugs (e.g., verapamil), as well as solute carrier (SLC) organic cation transporters. Collectively, these outcomes suggest that carrier-mediated transport systems, which differ from reported transporters and mechanisms, play a crucial role in the retinal distribution of amantadine across the inner/outer BRB.

## 1. Introduction

Retinal neurogenerative diseases, such as glaucoma and diabetic retinopathy, cause progressive visual deficit [1,2]. It is well-known that the progression of the visual deficit involves N-methyl-D-aspartate (NMDA) receptors [3,4]. Previous in vitro and in vivo analyses have indicated the involvement of overactivation of NMDA receptors in the loss of retinal ganglion cells (RGCs) [5,6], which transmit light stimuli from the eye to the brain. Recently, there has been an attempt to block NMDA receptors for the treatment of retinal diseases. For example, it has been reported that memantine, which is an adamantane derivative and an inhibitor of NMDA receptors, has been shown to exhibit neuroprotection in the retina of animal models of retinal diseases, both in vitro and in vivo [7,8]. Moreover, pharmacotherapy with memantine for glaucoma has reached phase III clinical trials [9]. Although a clinical trial of memantine failed [10], it is believed that adamantane derivatives have the potential to treat retinal diseases. Among the adamantane derivatives that are utilized for the blockage of NMDA receptors in clinical practice, amantadine shows the simplest structure. Therefore, scrutiny of the manner of retinal amantadine distribution could contribute to the clinical application of these derivatives.

The blood–retinal barrier (BRB) is known to regulate retinal drug distribution and is composed of retinal capillary endothelial and pigmented epithelial cells, termed the inner BRB and outer BRB, respectively [11]. Although the paracellular transport of compounds across these barriers is restricted by cellular tight junctions, recent studies have suggested that several ionic nutrients and drugs are supplied by blood-to-retina transport mediated by various plasma membrane transporters and transport systems [12]. As the pKa value of amantadine is 10.1, it is indicated that amantadine exists in the cationic form under physiological conditions (pH ~7.4) [13]. Accordingly, to develop a suitable strategy for efficient retinal amantadine distribution, an understanding of the retinal distribution of amantadine across the BRB would be valuable, in addition to clarifying the involvement of plasma membrane transport systems during this process.

To date, several solute carrier (SLC) transporters have been identified in amantadine transport in vivo and in vitro. As shown in Table 1, the roles of multidrug and toxin extrusion protein 1 (MATE1/Slc47a1) and neutral and basic amino acid transporter (ATB^0,+^/Slc6a14) at several tissues in amantadine transport have been reported [14,15]. Moreover, amantadine is reportedly a substrate for SLC organic cation transporter 1, abbreviated as OCT1 (Slc22a1), and OCT2 (Slc22a2) [16,17]. Among these transporters, mRNA expression of MATE1 has been observed in the inner and outer BRB model cells [18,19]. In addition to MATE1, the expression of other organic cation transporter subtypes has been documented. For example, plasma membrane monoamine transporter (PMAT/slc29a4) and organic cation/L-carnitine transporters 1-2 (OCTN1-2/slc22a4-5) are expressed at the BRB [18,19]. Moreover, putative cationic drug transport systems that accept several cationic and lipophilic drugs, including verapamil, clonidine, and propranolol, are known to exist in the inner BRB [20,21,22,23]. In these previous reports, amantadine reportedly exhibited an inhibitory effect on the uptake of these cationic drugs in inner BRB model cells, namely TR-iBRB2 cells [24]. Based on these lines of evidence, it can be speculated that these transporters and putative transport systems at the inner and outer BRB participate in amantadine transport to the retina across the BRB.

The present study aimed to clarify the details underlying the retinal distribution of amantadine. Herein, we used the retinal uptake index (RUI) experiment to elucidate the role of carrier-mediated processes in the retinal transfer of amantadine. In addition, the properties of amantadine transport across the inner and outer BRB were determined by using TR-iBRB2 cells and a conditionally immortalized rat RPE cell line (RPE-J cells) [25].

## 2. Materials and Methods

### 2.1. Animals and Reagents

The animal experiments performed in this study were approved by the Animal Care Committee of the University of Toyama with the registration numbers of A2017PHA-6 and A2020PHA-1. Male Wistar/ST rats (approximately 200 g; Japan SLC, Hamamatsu, Japan) were maintained under controlled conditions (12/12 h dark/light cycle; temperature, ~23 °C; humidity, around 50%). Male rats present an advantage for the comparison of previous reports of in vivo retinal distribution, as quantitative data of compound distribution to the retina have been obtained by using male rats [26]. A total of 13 rats were used for the assessment of in vivo blood-to-retina transport (control, *n* = 5; co-administration of unlabeled amantadine, *n* = 4; co-administration of PAH, *n* = 4). Amantadine HCl, [^3^H]- ([^3^H]amantadine; 0.3 Ci/mmol) was obtained from Moravek (Brea, CA, USA). From American Radiolabeled Chemicals (St. Louis, MO, USA), *n*-[1-^14^C]butanol ([^14^C]*n*-butanol; 2.0 × 10^3^ μCi/mmol) and verapamil [N-methyl-^3^H] hydrochloride ([^3^H]verapamil; 80 Ci/mmol) were purchased. Unlabeled drugs and compounds used in this study were commercially available.

### 2.2. Assessment of In Vivo Blood-to-Retina Transport

Rats were anesthetized by an intraperitoneal injection of pentobarbital sodium at 50 mg/kg. Using the rats, intracarotid artery injection was performed by following the method given in previous manuscripts [20,21,22,26], and the details of the procedure are given in the Appendix A. The functional retinal compound distribution as a percentage of the in vivo retinal transfer of [^14^C]*n*-butanol (RUI) was determined by using Equation (1).
RUI (%) = ([^3^H]amantadine/[^14^C]*n*-butanol (in retina))/([^3^H]amantadine/[^14^C]*n*-butanol (in the injected solution)) × 100(1)

### 2.3. In Vitro Transport Study

In 10% fetal bovine serum (FBS; Merck, Darmstadt, Germany)-containing Dulbecco’s modified Eagle’s medium (DMEM; Nissui Pharmaceutical, Tokyo, Japan) with 20 mM NaHCO_3_, 0.19 mM benzylpenicillin potassium, and 0.14 mM streptomycin sulfate, TR-iBRB2 cells (passage number 32–60) were cultured according to previous reports [20,21,22,24]. As described previously [25,27], RPE-J cells (passage number 79-92) were cultured in DMEM with 4% FBS, 20 mM NaHCO_3_, and 0.1 mM non-essential amino acid (FUJIFILM Wako Pure Chemical, Osaka, Japan) containing 0.16 mM benzylpenicillin potassium and 0.17 mM streptomycin sulfate. In accordance with established protocols for the uptake study [20,21,22], the cells were seeded at a density of 1.0 × 10^5^ cells/well onto a collagen I-coated 24-well plate (BioCoat™ Collagen I Cellware, Corning, Corning, NY, USA) and cultured for 2 days, at 33 °C, under 5% CO_2_/air. Referring to the previous reports [20,21,22,23,28,29], we started the uptake reaction, and the details of the procedure are included in the Appendix A. 

The uptake activities for [^3^H]amantadine and [^3^H]verapamil were calculated as the cell/medium ratio, using Equation (2).
Cell/medium ratio = [^3^H]compound (dpm) per cell protein (mg)/[^3^H]compound (dpm) per medium (µL)(2)

Kinetic parameters for cell uptake, such as the maximal uptake rate (*V*_max_), Michaelis–Menten constant (*K*_m_), and non-saturable uptake clearance (*K*_d_), were obtained by using the nonlinear least-squares regression analysis program (MULTI) [30] with Equations (3)–(5), where the uptake rate of the test compound and its concentration were *V* and *S*, respectively.
*V* = (*V*_max_ × *S*)*/*(*K*_m_ + *S*)(3)
*V* = (*V*_max_ × *S*)*/*(*K*_m_ + *S*) + *K*_d_ × *S*(4)
*V* = (*V*_max1_ × *S*)*/*(*K*_m1_ + *S*) + (*V*_max2_ × *S*)*/*(*K*_m2_ + *S*)(5)

### 2.4. Data and Statistical Analyses

All the data are shown as the mean ± standard deviation (SD). Using the unpaired two-tailed Student’s *t*-test (two groups) or one-way analysis of variance followed by Dunnett’s test (more than two groups), statistical differences were evaluated.

## 3. Results

### 3.1. In Vivo Blood-to-Retina Transport of [^3^H]Amantadine across the BRB

The in vivo RUI was observed to be 129 ± 13% (Figure 1), indicating that retinal distribution of [^3^H]amantadine is 1.29-fold greater than that of [^14^C]*n*-butanol. Following co-administration of 50 mM unlabeled amantadine, the RUI value significantly decreased by 44%. To test the effect of compounds at the same concentration, co-administration of 50 mM *p*-aminohippuric acid (PAH), which is an anionic compound that is reported to have no effect on the retinal distribution of cationic drugs, such as propranolol and clonidine [21,22], was performed. As a result, no significant effect was observed in the presence of 50 mM PAH (Figure 1).

### 3.2. Amantadine Uptake Properties in TR-iBRB2 Cells

TR-iBRB2 cells showed a time-dependent increase in [^3^H]amantadine uptake for at least 5 min. The initial uptake rate was presented as 14.0 ± 4.1 µL/(min·mg protein) (Figure 2A). At 4 °C, the [^3^H]amantadine uptake significantly reduced by 86% (Figure 2A). Concentration-dependent uptake of amantadine by TR-iBRB2 cells exhibited saturable and non-saturable processes, with a *V*_max_ of 1.36 ± 0.38 nmol/(min·mg protein), a *K*_m_ of 79.4 ± 27.3 µM, and a *K*_d_ of 2.75 ± 0.58 µL/(min·mg protein) (Figure 2B).

Furthermore, Na^+^-free and K^+^-replacement buffer had no significant effect on the uptake of [^3^H]amantadine by TR-iBRB2 cells; however, Cl^−^-free buffer significantly reduced the uptake of [^3^H]amantadine by 38% (Figure 3A). At an extracellular pH of 6.4, [^3^H]amantadine uptake by TR-iBRB2 cells was significantly reduced by 64% (Figure 3B). At an extracellular pH of 8.4, we noted the tendency of an increase in uptake of [^3^H]amantadine by TR-iBRB2 cells (Figure 3B). A proton ionophore, namely carbonyl cyanide-*p*-trifluoromethoxyphenylhydrazone (FCCP) [21,27], significantly decreased uptake by 50% at 50 μM (Figure 3C). In previous studies, it was indicated the acute treatment and pretreatment of NH_4_Cl caused alkalized and acidified intracellular pH in endothelial and epithelial cells, respectively [28,29]. Although acute NH_4_Cl treatment also induces the neutralization between the acid endosomal/lysosomal and intracellular compartments [27], we performed a study of [^3^H]amantadine uptake with acute and pretreatment of NH_4_Cl to examine the effect of H^+^-gradient on the intra- and extra-cellular compartments. The uptake was significantly decreased by 80% and increased by 90% at alkalized and acidified intercellular pH, respectively (Figure 3D). Under the experimental conditions in Figure 3, there is a possibility that the cytotoxicity may have affected the results of this uptake, because these experimental conditions are not ideal for the culture of these cells. However, the cellular protein amount after the uptake reaction was not significantly altered in each group compared with the control (Appendix A). Therefore, the cytotoxic effect caused by these experimental conditions is considered to be minimal.

### 3.3. Amantadine Uptake Properties in RPE-J Cells

In the current study, [^3^H]amantadine was time-dependently incorporated for at least 3 min in RPE-J cells, at an initial uptake rate of 15.9 ± 0.3 µL/(min·mg protein) (Figure 4A). At 4 °C, this uptake was significantly reduced by 87% (Figure 4A). Amantadine uptake by RPE-J cells involved both high- and low-affinity saturable processes (Figure 4B). Following the calculation, *K*_m1_ and *V*_max1_ values were 90.5 ± 49.6 µM and 0.914 ± 0.510 nmol/(min·mg protein), respectively, for the high-affinity uptake process. For the low-affinity uptake process, the *K*_m2_ and *V*_max2_ values were 9830 ± 1620 µM and 85.3 ± 11.4 nmol/(min·mg protein), respectively.

Na^+^-free, Cl^−^-free, and K^+^-replacement buffers had no significant impact on the uptake of [^3^H]amantadine by RPE-J cells (Figure 5A). The uptake of [^3^H]amantadine by RPE-J cells was significantly decreased by 32% and increased by 25% at extracellular pH of 6.4 and 8.4, respectively (Figure 5B). Furthermore, the uptake of [^3^H]amantadine by RPE-J cells was significantly decreased by 78% and increased by 18% at alkalized and acidified intercellular pH, respectively (Figure 5C). The cellular protein in each group of Figure 5 also had no significant difference compared with control (Appendix A).

### 3.4. Inhibition of [^3^H]Amantadine Transport by Drugs/Compounds

The inhibitory effects on the in vitro [^3^H]amantadine uptake are summarized in Table 2. At 0.2 mM, several cationic drugs (desipramine, imipramine, quinidine, and verapamil) strongly inhibited [^3^H]amantadine uptake by both TR-iBRB2 cells and RPE-J cells. 

Memantine, pyrilamine, amantadine, and clonidine (0.2 mM) exhibited a marked inhibition of [^3^H]amantadine uptake by TR-iBRB2 cells but not RPE-J cells; however, these cationic drugs significantly inhibited [^3^H]amantadine uptake by RPE-J cells at 1 mM. SLC organic cation/anion transporter substrates, such as PAH, cimetidine, choline, 1-methyl-4-phenylpyridinium (MPP^+^), and decynium-22, did not suppress [^3^H]amantadine uptake. Similarly, anionic and cationic amino acids, including L-glutamic acid and L-arginine, did not significantly impact [^3^H]amantadine uptake.

### 3.5. Mutual Effect of Amantadine and Verapamil on Uptake by TR-iBRB2 Cells

We next examined the involvement of verapamil-sensitive putative transport systems in the inner BRB [20] in amantadine transport. Accordingly, we performed a kinetic analysis of amantadine uptake by TR-iBRB2 cells in the presence of verapamil. On analyzing the Lineweaver–Burk plot, the fitted line of concentration-dependent amantadine uptake by TR-iBRB2 cells in the presence of 10 μM verapamil did not intersect with that in the absence of verapamil, on both the *y*-axis and *x*-axis (Figure 6A). In the presence of amantadine at 220 μM, the fitted line of concentration-dependent verapamil uptake intersected at the *y*-axis with that in the absence of amantadine (Figure 6B); this indicated that amantadine competitively inhibits verapamil-sensitive transport mechanisms at the inner BRB.

### 3.6. Effect of Amantadine on Concentration-Dependent Verapamil Uptake by RPE-J Cells

The effect of amantadine on verapamil-sensitive putative transport systems in the outer BRB was investigated. Herein, we investigated the time-dependent uptake of verapamil by RPE-J cells. RPE-J cells showed a time-dependent increase with an initial uptake rate of 70.4 ± 5.6 µL/(min·mg protein) in [^3^H]verapamil uptake for at least 3 min (Figure 7A). At 1 min, this [^3^H]verapamil uptake was reduced with 200 µM unlabeled verapamil by 86% (Figure 7A). In RPE-J cells, verapamil uptake exhibited a saturable process, with a *V*_max_ value of 6.31 ± 0.35 nmol/(min·mg protein) and a *K*_m_ value of 55.6 ± 5.2 µM (Figure 7B). In the Lineweaver–Burk plot analysis, the fitted line of concentration-dependent verapamil uptake by RPE-J cells in the presence of 200 μM amantadine did not intersect with that in the absence of verapamil, at both the *y*-axis and *x*-axis (Figure 7B); this suggested that amantadine does not competitively or non-competitively inhibit verapamil transport at the outer BRB.

## 4. Discussion

We assessed the inner and outer BRB-mediated transport of amantadine, as well as the carrier-mediated transport of amantadine in the inner and outer BRB. Detailed in vitro analyses clarified characteristics of amantadine transport mechanisms at the BRB; however, the involvement of typical organic cation transporters and putative cationic drug transport systems in the carrier-mediated amantadine transport at the BRB is not suggested.

The in vivo retinal drug transfer study (Figure 1) indicated that amantadine underwent active retinal distribution from the circulating blood. Previously, we reported the correlation between in vivo retinal distribution of drugs transported by passive diffusion and lipophilic properties, indicated as log D {RUI = 46.2 × exp(0.515 × log D)} [26]. The log D value of amantadine is 0.176 [31]. Therefore, if amantadine is only transported to the rat retina by passive diffusion across the in vivo BRB, the RUI value of amantadine can be estimated as 50.4% {= 46.2 × exp(0.515 × 0.176)}. Compared with this estimated value, the RUI value of [^3^H]amantadine in the present study was 2.5-fold greater (129%). In addition, unlabeled amantadine at 50 mM showed a significant reduction in the [^3^H]amantadine RUI; however, PAH at the same concentration demonstrated a minimal impact. Based on these findings, it can be suggested that carrier-mediated transport at the inner and outer BRB promotes the retinal distribution of amantadine.

Furthermore, transport studies using in vitro models of the inner and outer BRB detected carrier-mediated amantadine transport at these barriers. In TR-iBRB2 cells, both saturable and non-saturable components were involved in the uptake of amantadine, with a *K*_m_ value of 79.4 µM (Figure 2B). As the contribution ratio of the saturable component was 86% {= *V*_max_/*K*_m_ ÷ (*V*_max_/*K*_m_ + *K*_d_) × 100}, carrier-mediated processes of amantadine uptake could play an important role in amantadine transport at the inner BRB. In terms of the kinetic analysis of amantadine uptake by RPE-J cells, it was observed that this uptake was composed of two saturable processes, with *K*_m_ values of 90.5 and 9830 µM for high- and low-affinity processes, respectively (Figure 4B). The contributions of high- and low-affinity processes were calculated as 54% {= *V*_max1_/*K*_m1_ ÷ (V_max1_/*K*_m1_ + *V*_max2_/*K*_m2_) × 100} and 46% {= *V*_max2_/*K*_m2_ ÷ (*V*_max1_/*K*_m1_ + *V*_max2_/*K*_m2_) × 100}, respectively; this indicated that the involvement of the high-affinity process in carrier-mediated amantadine transport at the outer BRB is equal to that of the low-affinity process. Collectively, these results suggest that carrier-mediated processes play a major role in amantadine transport across the inner and outer BRB.

As described in the previous section, the *K*_m_ value for the saturable process of amantadine uptake by TR-iBRB2 cells (79.4 µM) is similar to that determined for the high-affinity process of amantadine uptake by RPE-J cells (90.5 µM). However, the contribution ratios of saturable kinetics (50–100 μM) in TR-iBRB2 and RPE-J cells were 86% and 54%, respectively. This difference may influence the properties of net amantadine transport at the inner and outer BRB. For example, TR-iBRB2 cells exhibited Cl^−^-sensitive [^3^H]amantadine uptake; this effect was not observed in RPE-J cells (Figure 3A and Figure 5A). Moreover, several cationic compounds, such as memantine, pyrilamine, and clonidine, demonstrated a lower inhibitory effect (0.2 mM) on [^3^H]amantadine uptake by RPE-J cells than that on uptake by TR-iBRB2 cells (Table 2). The characteristics of high- and low-affinity processes need to be individually evaluated to clarify the differences. Nevertheless, our study, including kinetic analyses, implies that the net transport of amantadine at the inner BRB differs from that at the outer BRB.

Although several transporters reportedly accept amantadine as a substrate, our study indicates a minor contribution of these transporters and putative transport systems in terms of amantadine transport at the inner and outer BRB. We observed no inhibitory effect of OCTs and/or ATB^0,+^ inhibitors, such as decynium-22, TEA, MPP^+^, glycine, and L-arginine, on [^3^H]amantadine uptake by both model cells (Table 2) [32,33,34,35,36,37,38]; this indicates that OCTs and ATB^0,+^ were not involved in amantadine transport at the inner and outer BRB. In addition, typical organic cation transporters that are expressed in the inner and outer BRB, such as OCTN1-2, MATE1, and PMAT, play a minor role in amantadine transport because substrates of these transporters (cimetidine, L-carnitine, TEA, MPP^+^, and serotonin [39,40,41,42,43,44,45,46]) did not significantly reduce [^3^H]amantadine uptake by these cells (Table 2). Regarding unidentified cationic drug transport systems for clonidine, propranolol, and verapamil at the BRB [20,21,22], amantadine transport mechanisms at the inner and outer BRB appear distinct from these cationic drug transport systems based on our results. The *K*_m_ values for clonidine and propranolol uptake by TR-iBRB2 cells are 286 and 237 μM, respectively [21,22]. As [^3^H]amantadine uptake by TR-iBRB2 cells was reduced by more than 58% in the presence of clonidine and propranolol (0.2 mM), which is lower than the *K*_m_ values described above, it is unlikely that putative cationic drug transport systems for clonidine and propranolol participate in amantadine transport at the inner BRB. Reportedly, novel verapamil transport systems have been identified in the rat inner BRB in vitro [20]. In addition, the uptake study of verapamil using RPE-J cells indicated the existence of carrier-mediated verapamil transport in the rat outer BRB (Figure 7), as well as human RPE cells [23]. However, it is strongly suggested that amantadine transport systems at the inner and outer BRB are distinct from the verapamil transport systems considering the mutual effect of amantadine and verapamil on uptake by TR-iBRB2 and RPE-J cells (Figure 6A and Figure 7B). Collectively, the results of the functional study suggest that retinal distribution of amantadine across the inner and outer BRB occurs via carrier-mediated transport systems that do not consist of known organic cation transporters and putative organic cation transport systems for cationic drugs, including verapamil.

Parsons et al. reported that amantadine inhibits NMDA receptors, with an IC_50_ of 20 µM [47]. During pharmacotherapy for Parkinson’s disease, the plasma concentration range of 0.6–29 μM for amantadine has been established [48]. As the *K*_m_ values for the relatively high-affinity process of amantadine transport in the in vitro inner and outer BRB model cells (50–100 μM) were greater than the concentrations related to the pharmacology and pharmaceutics of amantadine, carrier-mediated amantadine transport at the inner/outer BRB is critical for understanding retinal amantadine distribution. Hence, identifying amantadine transport systems at the BRB could help establish an appropriate pharmaceutical strategy for amantadine applications in retinal diseases. For the identification of the molecule involved in the relatively high-affinity process of amantadine uptake, TR-iBRB2 cells may afford an advantage, as the relatively high-affinity process was the only observed carrier-mediated process of amantadine transport.

## 5. Conclusions

In the current study, we demonstrated the process of retinal amantadine transport. The RUI experiment indicated the retinal distribution of amantadine. Moreover, the uptake study using in vitro model cells suggested the involvement of transport systems in amantadine blood-to-retina transport across the inner and outer BRB. Furthermore, it can be suggested that transport systems for amantadine at the inner/outer BRB are independent of cationic drug transport systems for verapamil, as well as SLC organic cation and amino acid transporters. These transport systems for amantadine at the BRB promote amantadine distribution to the retina. Therefore, the characteristics of amantadine transport at the BRB identified in this study can provide an in-depth understanding of amantadine-sensitive transport mechanisms at the BRB and, thus, the utilization of adamantane derivatives, such as amantadine, for retinal diseases.

## Figures and Tables

**Figure 1 pharmaceutics-13-01339-f001:**
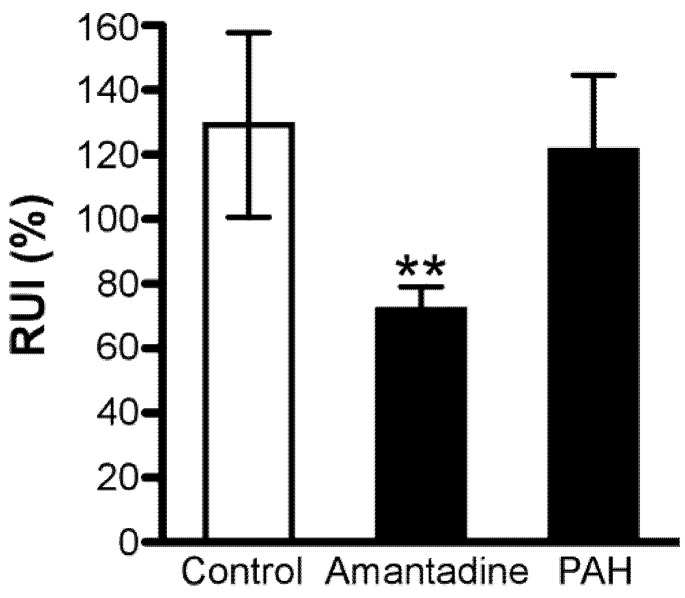
[^3^H]Amantadine RUI in rats. [^3^H]Amantadine (5 μCi/rat) and [^14^C]*n*−butanol (0.5 μCi/rat) were injected in the absence (control, *n* = 5) or presence of 50 mM unlabeled amantadine (*n* = 4) or 50 mM PAH (*n* = 4). Each column, expressing [^3^H]amantadine RUI, represents the mean ± SD; ** *p* < 0.01, significant difference from control. RUI, retinal uptake index; PAH, *p*−aminohippuric acid.

**Figure 2 pharmaceutics-13-01339-f002:**
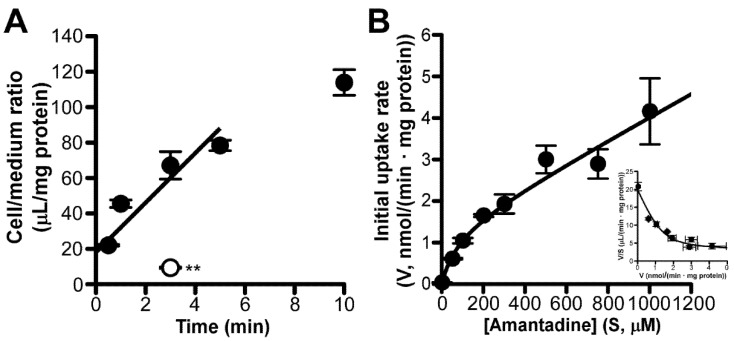
Time-, temperature−, and concentration−dependent uptake of [^3^H]amantadine by TR−iBRB2 cells. (**A**) Time dependency of [^3^H]amantadine uptake (1.67 µM, 0.1 µCi/well) by TR−iBRB2 cells at 37 °C (control; closed circles) and effect of low temperature (4 °C; open circle) on the uptake; ** *p* < 0.01, significant difference from the control. (**B**) Concentration dependency of amantadine uptake by TR−iBRB2 cells. Amantadine uptake was examined at 37 °C for 3 min over the concentration range of 1.67 to 1000 µM and analyzed by using Michaelis–Menten and Eadie–Scatchard (inset) plots. Each point in the figure represents the mean ± SD (*n* = 3).

**Figure 3 pharmaceutics-13-01339-f003:**
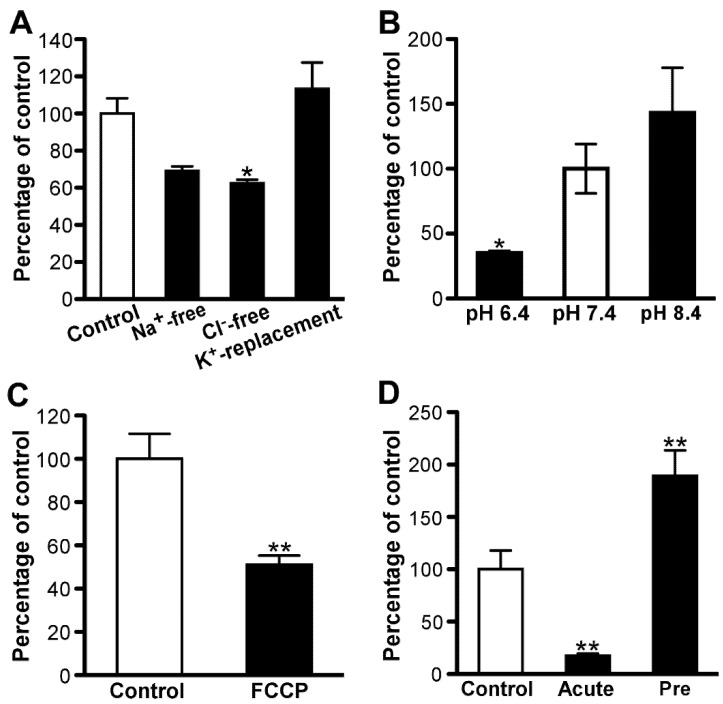
Uptake properties of [^3^H]amantadine by TR−iBRB2 cells. (**A**) Effect of Na^+^, Cl^−^, and membrane potential on [^3^H]amantadine uptake (1.67 µM, 0.1 µCi/well) for 3 min at 37 °C. (**B**) Effect of extracellular pH on [^3^H]amantadine uptake by TR−iBRB2 cells. (**C**) Effect of carbonyl cyanide−*p*−trifluoromethoxyphenylhydrazone (FCCP) treatment at 50 µM on [^3^H]amantadine uptake. (**D**) Effect of intracellular pH on the uptake of [^3^H]amantadine for 15 s by TR−iBRB2 cells at 37 °C. Pretreatment (Pre) and acute treatment with 30 mM NH_4_Cl were performed to decrease and increase intracellular pH, respectively. Each column in the figure represents the mean ± SD (*n* = 3); * *p* < 0.05, ** *p* < 0.01, significantly different from the control.

**Figure 4 pharmaceutics-13-01339-f004:**
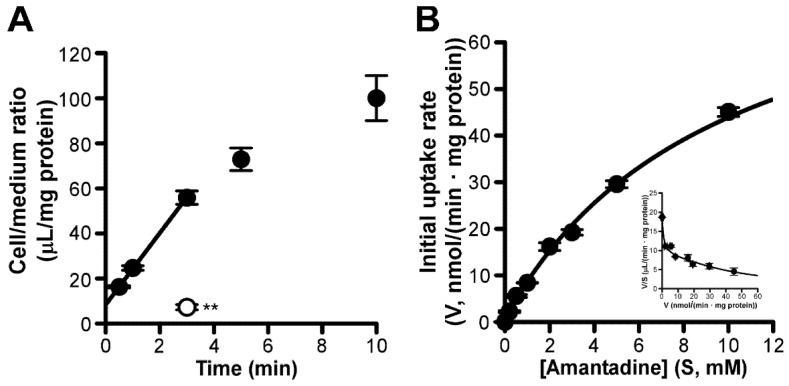
Time−, temperature−, and concentration−dependent uptake of [^3^H]amantadine by RPE−J cells. (**A**) Time dependency of [^3^H]amantadine uptake (1.67 µM, 0.1 µCi/well) by RPE−J cells (37 °C; control, closed circles) and effect of low temperature (4 °C; open circle) on [^3^H]amantadine uptake; ** *p* < 0.01, significant difference from the control. (**B**) Concentration−dependent uptake of amantadine (1.67 µM−10 mM) by RPE-J cells was expressed by using Michaelis–Menten and Eadie–Scatchard (inset) plots. Each point in the figure represents the mean ± SD (*n* = 3).

**Figure 5 pharmaceutics-13-01339-f005:**
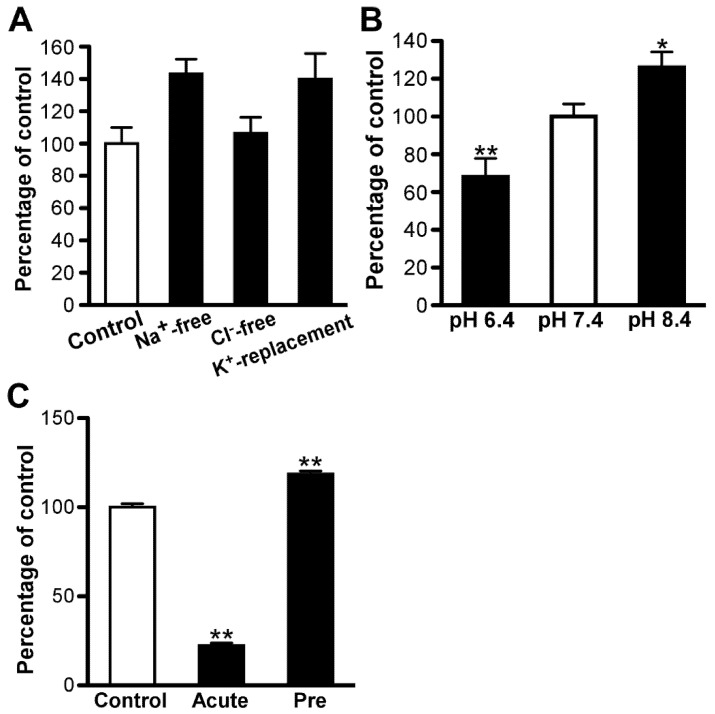
Properties of [^3^H]amantadine uptake by RPE−J cells. (**A**) Effect of Na^+^, Cl^−^, and membrane potential on [^3^H]amantadine uptake (1.67 µM, 0.1 µCi/well) for 3 min at 37 °C. (**B**) Effect of extracellular pH on the uptake of [^3^H]amantadine. (**C**) Effect of intracellular pH on [^3^H]amantadine uptake for 15 s at 37 °C by RPE−J cells. Pretreatment (Pre) and acute treatment with 30 mM NH_4_Cl were performed to decrease and increase intracellular pH, respectively. Each column in the figure represents the mean ± SD (*n* = 3); * *p* < 0.05, ** *p* < 0.01, significantly different from the control.

**Figure 6 pharmaceutics-13-01339-f006:**
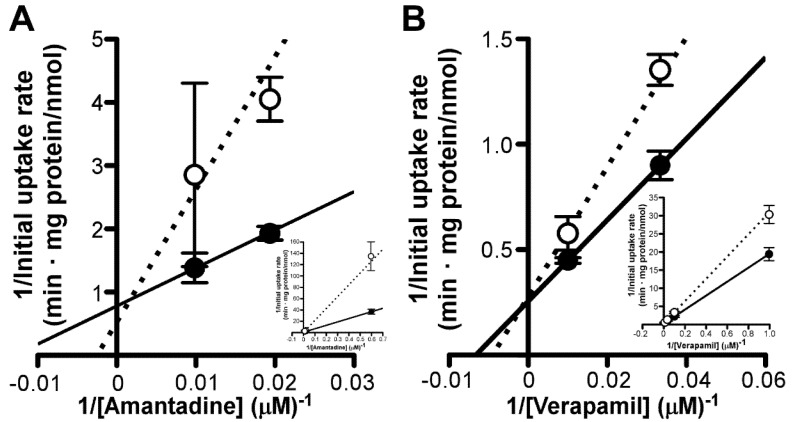
Mutual effect on uptake by TR−iBRB2 cells of amantadine (**A**) and verapamil (**B**). (**A**) The uptake of amantadine at a concentration of 1.67, 50, 100, and 300 µM was assessed at 37 °C for 3 min, with (open circles, dotted line) or without (closed circles, solid line) 10 µM verapamil. (**B**) The uptake of verapamil at a concentration of 1, 10, 100, and 300 µM was examined at 37 °C for 3 min, with (open circles, dotted line) or without (closed circles, solid line) 220 µM amantadine. Each point in the Lineweaver–Burk plot of all data (inset) and highlighted data at the high concentration range represents the mean ± SD (*n* = 3).

**Figure 7 pharmaceutics-13-01339-f007:**
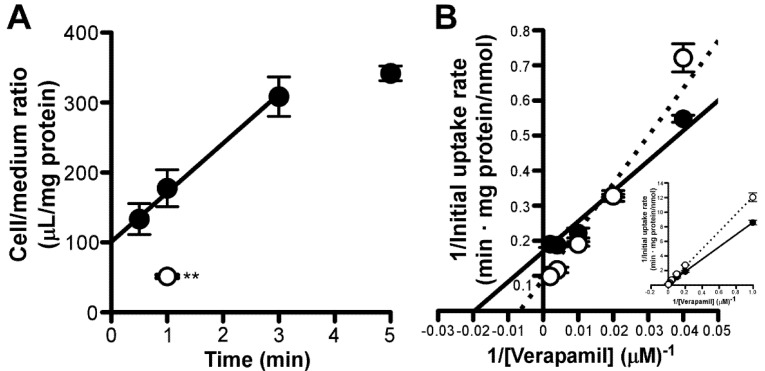
Effects of amantadine on uptake of [^3^H]verapamil by RPE−J cells. (**A**) Time dependency and self-inhibition of [^3^H]verapamil uptake by RPE−J cells. RPE−J cells were incubated with [^3^H]verapamil (6.25 nM, 0.1 µCi/well), in the presence (open circle) or absence (control, closed circles) of 200 µM verapamil at 37 °C. (**B**) Lineweaver–Burk plot of all data (inset) and highlighted data at the concentration of 5, 10, 25, 50, 100, 250, and 500 µM verapamil uptake at 37 °C for 3 min, in the absence (closed circles, solid line) or presence (open circles, dotted line) of 100 µM amantadine by RPE−J cells. Each point in the Figure represents the mean ± SD (*n* = 3); *** p* < 0.01, significant difference from the control.

**Table 1 pharmaceutics-13-01339-t001:** Candidates of amantadine transporter and transport system in the retina.

Name		Is Amantadine Accepted as a Substrate?	Is It Expressed in the Retina?
Slc6a14	ATB^0,+^	Yes [15]	N.D.
Slc22a1	OCT1	Yes [16]	N.D.
Slc22a2	OCT2	Yes [17]	N.D.
Slc22a3	OCT3	N.D.	N.D.
Slc22a4	OCTN1	N.D.	Yes (mRNA) [18]
Slc22a5	OCTN2	N.D.	Yes (mRNA) [18]
Slc29a4	PMAT	N.D.	Yes (mRNA) [19]
Slc47a1	MATE1	Yes [14]	Yes (mRNA) [19]
Putative transport systems	Verapamil	N.D.	Yes [20,21,22,23]
	Clonidine		
	Propranolol		

N.D., not determined.

**Table 2 pharmaceutics-13-01339-t002:** Effect of compounds on the uptake of [^3^H]amantadine by TR−iBRB2 cells and RPE−J cells.

Compound	Concentration	% of Control	
	(mM)	TR-iBRB2	RPE-J
Control		100 ± 12	100 ± 12
Desipramine	0.2	7.05 ± 0.57 **	10.7 ± 5.6 **
Imipramine	0.2	8.37 ± 1.43 **	16.9 ± 5.2 **
Propranolol	0.2	8.61 ± 0.75 **	15.7 ± 5.9 **
Quinidine	0.2	13.7 ± 1.8 **	34.1 ± 3.0 **
Memantine	0.2	16.7 ± 2.2 **	123 ± 38 **
	1.0	N.D.	27.1 ± 2.6 **
Pyrilamine	0.2	20.9 ± 1.8 **	77.8 ± 38.4
	1.0	N.D.	18.2 ± 1.7 **
Verapamil	0.2	22.1 ± 6.2 **	20.5 ± 4.8 **
Amantadine	0.2	34.1 ± 4.1 **	89.9 ± 5.6
	1.0	N.D.	21.3 ± 3.1 **
Timolol	0.2	34.6 ± 10.6 **	59.1 ± 5.1
Clonidine	0.2	42.1 ± 12.1 **	71.6 ± 4.7
	1.0	N.D.	30.8 ± 4.0 **
Pyrimethamine	0.2	44.3 ± 8.8 **	88.9 ± 3.3
PAH	0.2	85.8 ± 10.8	139 ± 33
Acetazolamide	0.2	90.9 ± 3.8	N.D.
Gluconate	0.2	91.7 ± 21.7	86.2 ± 23.4
Cimetidine	0.2	97.6 ± 24.7	106 ± 5
Choline	0.2	101 ± 8	124 ± 39
MPP^+^	0.2	106 ± 31	75.5 ± 5.0
	1.0	N.D.	91.9 ± 5.6
Decynium-22	0.2	106 ± 31	86.7 ± 41.9
	0.5	N.D.	136 ± 8
L-Carnitine	0.2	111 ± 35	95.1 ± 5.2
	2.5	99.5 ± 2.3	119 ± 28
TEA	0.2	117 ± 45	87.7 ± 11.8
	1.0	N.D.	81.8 ± 10.3
Serotonin	0.2	126 ± 60	158 ± 52 **
L-Glutamic acid	2.5	80.2 ± 10.0	84.5 ± 16.4
L-Aspartic acid	2.5	101 ± 23	79.5 ± 12.7
Glycine	2.5	102 ± 4	116 ± 13
L-Leucine	2.5	112 ± 17	113 ± 16
L-Arginine	2.5	127 ± 15	150 ± 38 **

Uptake of [^3^H]amantadine (1.67 µM, 0.1 µCi/well) by the indicated cells was performed for 3 min at 37 °C. Each value represents the mean ± SD (*n* = 3–30); *** p* < 0.01, significantly different from the control. PAH, *p*−aminohippuric acid; MPP^+^, 1−methyl−4−phenylpyridinium; TEA, tetraethylammonium; N.D., not determined.

## Data Availability

The data of this study are available from the corresponding author upon reasonable request.

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
