# Peer review of "Comprehensive Evidence of Carrier-Mediated Distribution of Amantadine to the Retina across the Blood–Retinal Barrier in Rats"

_pharmaceutics, 2021, doi:10.3390/pharmaceutics13091339_

Round 1
Reviewer 1 Report
Authors investigated in-vivo and in-vitro the amantadine up-take by blood-retinal-barrier (BRB). Additionally, authors have investigated amantadine-drug interactions regarding amantadine uptake by BRB. Authors identified that amantadine likely interact with the same carrier of verapamil. Overall the manuscript is well written and has scientific soundness, however it needs some revisions.
Introduction
- as regards as neuroprotection in glaucoma cite the following papers PMID: 28925883
- correctly refere to the history of memantine clinical trial failure in glaucoma
Methods:
- clearly indicate the number of animals per experimental group. Report also in the animal section of the methods the protocol number provided by the ethical commitee
- authors reported data as mean+/-SEM but this is not correct, since most of experiments were carried out on 3 or 4 biological replicates. SEM should be used for n>7.
Results:
- figure 1. authors must report the number of animals per experimental group
- figure 2-7. SEM cannot be reported for 3 biological replicates, authors must report SD
- figure 6. Authors reported a Lineweaver-Burk plot with only two points per line, this is not correct, a third point is necessary, since a linear regression with two points is meaningless. In fact, in figure 7 authors reported Lineweaver-Burk plots with more than 3 points.
Discussion:
- The sentence "Adamantane derivatives, including amantadine, are expected to be developed as therapeutic agents for retinal diseases such as glaucoma" is quite trivial, and should be rephrased, e.g. reporting any reference, regarding this expectations. Instead, authors should hypothesize that adamantane derivatives would be interesting ligands for development of neuroprotective drugs for treatment of retinal diseases such as glaucoma.
- authors should discuss about the hypothetic route of administration of adamantane derivatives for treatment of retinal degenerative diseases.
Author Response
Response to Reviewer #1
Introduction: As regard as neuroprotection in glaucoma cite the following papers PMID: 28925883 correctly refer to the history of memantine clinical trial failure in glaucoma.
Following your kind comment, the clinical trial failure in glaucoma of memantine was clearly described in the revised manuscript with referring the suggested review. We are grateful if you check the revision in first paragraph of the Introduction section.
Methods: Clearly indicate the number of animals per experimental group. Report also in the animal section of the methods the protocol number provided by the ethical committee.
Authors reported data as mean+/-SEM but this is not correct, since most of experiments were carried out 3 or 4 biological replicates. SEM should be used for n>7.
Results: figure 1. authors must report the number of animals per experimental group. figure 2-7. SEM cannot be reported for 3 biological replicates, authors must report SD.
Thank you for your suggestion. Although the approval of animal study was described in the “Institutional Review Board Statement” section, the descriptions are also included in the “Materials and Methods” section. In the “Materials and Methods” section, we also added the information of the total animal number used in this study. The animal number in each group has already described in the legend for Figure 1.
We also agree with the use of standard deviation (S.D.). All error bars in our manuscript are changed to be explained as S.D..
Authors reported a Lineweaver-Burk plot with only two points per line, this is not correct, a third point is necessary, since a liner regression with points is meaningless. In fact, in figure 7 authors reported Lineweaver-Burk plot more than 3 points.
Thank you for your critical comment. To increase the reliability of the Lineweaver-Burk plots, the main panel of these figures are revised with showing at least 3 points of the data. Moreover, the all concentration points are clearly shown in the legend of Figures 6 and 7. We are grateful if you satisfy the revision of the Figures and descriptions of the manuscript.
The sentence “Adamantane derivatives, including amantadine, are expected to be developed as therapeutic agents for retinal diseases such as glaucoma.” is quite trivial, and should be rephrased, e.g. reporting any reference, regarding this expectations. Instead, authors should hypothesize that adamantane derivatives would be interesting ligands for development of neuroprotective drugs for treatment of retinal diseases such as glaucoma.
Thank you for your kind comment. As Reviewer #4 commented, the discussion section is hard to follow. Therefore, we have tried to reduce the descriptions in the Discussion section. To re-check the last paragraph including the sentence which the reviewer pointed out, it is considered that this sentence yields to the redundancy of the Discussion. Moreover, the similar sentence is also existed in the “Conclusion” section. Taking the above points into consideration, this sentence in the Discussion section is deleted in the revised manuscript. It would be grateful if the reviewer is able to agree with our idea.

Reviewer 2 Report
The manuscript by Shinozaki et al. describes a carrier-mediated retinal distribution of amantadine across the blood-retinal barrier. In their study the authors assessed transport of amantadine in vivo and in in vitro using model cells of the inner and outer BRB. The authors conclude that the carrier-mediated transport systems play a crucial role in the retinal distribution of amantadine across the inner/outer BRB. This results are interesting and novel, while the manuscript is well written and easy to follow. Thus I am happy to recommend this article for the publication in its current state.
Author Response
Response to Reviewer #2
The manuscript by Shinozaki et al. describes a carrier-mediated retinal distribution of amantadine across the blood-retinal barrier. In their study the authors assessed transport of amantadine in vivo and in in vitro using model cells of the inner and outer BRB. The authors conclude that the carrier-mediated transport systems play a crucial role in the retinal distribution of amantadine across the inner/outer BRB. These results are interesting and novel, while the manuscript is well written and easy to follow. Thus, I am happy to recommend this article for the publication in its current state.
Thank you for the comment. Following the comments from the other reviewers, we have additionally improved the manuscript. It is expected that this revision increases a quality of our manuscript. We are grateful if you also satisfy this revised version.

Reviewer 3 Report
The present manuscript studied the distribution of Amantadine across blood retinal berrier using both in vivo and in vitro models. The authors showed involvement of carrier mediated transport in the retinal distribution of amantadine.
My comments
1- The title should be changed to show that it is an experimental study.
2- The authors started the introduction focusing on glaucoma, although many retinal diseases can affect BRB with involvement of NMDA such as diabetic retinopathy. Therefore, it is better not to focus on certain retinal disease.
3-Why only male rats were used in the study?
4- A graphical abstract showing the structure of amantadine along with the reported distribution among inner and outer BRB is recommended
Author Response
Response to Reviewer #3
- The title should be changed to show that it is an experimental study.
Thank you for your helpful recommendation. We have changed the title of our manuscript for the readers to understand our study as the experimental one.
- The authors started the introduction focusing on glaucoma, although many retinal diseases can affect BRB with involvement of NMDA such as diabetic retinopathy. Therefore, it is better not to focus on certain retinal disease.
Thank you for giving us the valuable comment. As the reviewer pointed out, NMDA receptor-related signaling is also related to various retinal diseases. We have changed first paragraph of Introduction not to focus on the glaucoma. We hope that you satisfy the revised Introduction.
- Why only male rats were used in this study?
It is because the quantitative data of compound/drug distribution to the retina have been obtained by using male rats. For example, the relationship between the in vivo retinal distribution of compounds/drugs and their lipophilicity was tested by using male rats [1]. Like this, the use of male rats has an advantage of comparing the previous outcomes of in vivo retinal distribution. Following your valuable comment, we added the above information to the Materials and Methods section for the readers to understand this advantage.
- A graphical abstract showing the structure of amantadine along with the reported distribution among inner and outer BRB is recommended.
Thank you for your valuable comment. We added a graphical abstract of the summary of our study with the amantadine structure.
References
- Hosoya, K.; Yamamoto, A.; Akanuma, S.; Tachikawa, M. Lipophilicity and transporter influence on blood-retinal barrier permeability: a comparison with blood–brain barrier permeability. Res. 2010, 27, 2715-2724.

Reviewer 4 Report
In this manuscript, the authors investigated the retinal transport of amantadine across the inner and outer blood-retinal barrier (BRB) using in vivo and in vitro models. The presented results are potentially significant for the future clinical use of amantadine for treating retinal diseases such as glaucoma. However, the experiments, data presentation, and interpretation need further clarifications. Also, the studies mainly use traditional biochemical approaches, but the authors need to demonstrate experiments focusing on mechanistic aspects about the carrier-mediated delivery of amantadine into the eyes. The detailed comments are listed below.
- The authors can improve the introduction—the research background and focus can be more clearly summarized. In this manner, the introduction part can be more valuable to the research community.
- The introduction is a bit hard to follow, especially for non-experts. A schematic explanation or a summary table might be helpful to understand the research background and the objective.
- In line 131, the authors described that “the in vivo retinal distribution of [3H]amantadine was examined using the RUI method and was 129% (Fig 1). Why is this over 100%? Based on the equation, this point needs further clarifications. Also, PAH was a negative control. Its rationale is missing.
- In lines 160-161, the authors described that "uptake was significantly decreased by 80% and increased by 90% at alkalized and acidified intercellular pH, respectively (Fig 3D)". The authors used NH4Cl to neutralize acidic endosomal-lysosomal compartments, but the rationale is missing in the text. Also, the authors need to explain what the pretreatment and acute treatment are in Fig 3. Otherwise, readers cannot understand the rationale of the experiment.
- In Fig 3, the authors investigated the effects of different ions or pH on uptake of amantadine by TR-iBEB2 cells. Those conditions are not ideal for cell culture. The authors need to assess the cell viability and cytotoxicity of the treatments to conclude.
- The discussion section is currently a bit too descriptive and hard to follow. This section can be more focused.
Author Response
Response to Reviewer #4
- The authors can improve the introduction the research background and focus can be more clearly summarized. In this manner, the introduction part can be more valuable to the research community.
Thank you for your beneficial comment. We are able to agree with your concern, and have revised the Introduction section concisely. Especially, the fourth paragraph of the Introduction section was drastically modified since it is considered that this part contains the redundant descriptions with additional Table 1. It would be grateful if you satisfy this revision of the Introduction section.
- The introduction is bit hard to follow, especially for nonexperts. A schematic explanation or a summary table might be helpful to understand the research background and the objective.
Thank you for your helpful recommendation. We added the graphical abstract for the readers to understand the structure of the inner/outer BRB easily. Moreover, as described in the above comment, Table which include the previous information of candidates for amantadine transport was additionally included. We hope that this information is helpful for readers to follow introduction.
- In line 131, the authors described that “the in vivo retinal distribution of [3H]amantadine was examined using the RUI method and was 129% (Fig 1).” Why is this over 100%? Based on the equation, this point needs further clarifications. Also, PAH was a negative control. Its rational is missing.
Thank you for your recommendation. As described in the manuscript, this percentage indicates the retinal transfer of the test compound relative to that of [14C]n-butanol. Hence, this value as 129% means that retinal distribution of amantadine is 1.29-fold greater than that of n-butanol. [14C]n-Butanol is known as a highly-diffusible compound to the retina, but it has been reported that several compounds, such as verapamil and propranolol, exhibit promotive retinal transfer compared with n-butanol and more than 100% as the RUI value [1-3]. In the revised manuscript, the explanation of the above points is included in the Results section.
In the inhibition study, to examine the effect of compounds at the same concentrations of unlabeled amantadine (50 mM), the co-administration of PAH was also performed. PAH is an anionic compound and is reported to show no effect on the retinal distribution of other cationic compounds, such as verapamil [2,3]. To explain these points more clearly, we added the text in the result section.
- In the lines 160-161, the authors described that “uptake was significantly decreased by 80% and increased by 90% at alkalized and acidified intercellular pH respectively (Fig 3D).” The authors used NH4Cl to neutralize acidic endosomal-lysosomal compartments, but the rationale is missing in the text. Also, the authors need to explain what the pretreatment and acute treatment are in Fig 3. Otherwise, readers cannot understand the rational of the experiment.
As the reviewer pointed out, acute NH4Cl treatment also causes the neutralization between acidic endosomes/lysosomes and the intracellular compartment. It has not been reported that amantadine is trapped into the acidic endosomes and lysosomes. So, amantadine trapping into the acidic endosomal-lysosomal compartments needs to be tested for understanding the detail of amantadine transport under NH4Cl treatment at the inner and outer BRB in further. Nevertheless, the previous reports have indicated that the change in intracellular pH by acute treatment and pre-treatment of NH4Cl is exhibited in endothelial and epithelial cells [4,5]. Hence, it is implied that H+-gradient between the intra- and extra-cellular compartments affects amantadine uptake by TR-iBRB2 and RPE-J cells, at least in part.
In the revised manuscript, the above points are included in the Results section. In addition, the explanation of the compound name, NH4Cl, was added to the Results section. Thank you for your valuable comment.
- In Fig 3, the authors investigated the effects of different ions or pH on uptake of amantadine by TR-iBRB2 cells. Those conditions are not ideal for cell culture. The authors need to assess the cell viability and cytotoxicity of the treatments to conclude.
Thank you for your valuable comment. In this study, specific analyses to assess the cellular viability and cytotoxicity have not been performed. However, in this study, we monitored the protein amount derived from the cells attached onto the well. There is no significant difference of cellular protein in the groups of Figure 3, indicating that the cytotoxic effect of these not ideal conditions is minimum in TR-iBRB2 and RPE-J cells. In the revised manuscript, clear descriptions of the cellular protein amounts are added in the paragraphs which explain the results of Figure 3.
- The discussion section is currently a bit too descriptive and hard to follow. This section can be more focused.
Thank you for your beneficial comment. In the revised manuscript, redundant and descriptive descriptions in the Discussion section were minimized. We are grateful if the reviewer is able to accept the revised version of the Discussion.
Finally, thank you very much again for your valuable and helpful comments.
References
- Hosoya, K.; Yamamoto, A.; Akanuma, S.; Tachikawa, M. Lipophilicity and transporter influence on blood-retinal barrier permeability: a comparison with blood–brain barrier permeability. Res. 2010, 27, 2715-2724.
- Kubo, Y.; Kusagawa, Y.; Tachikawa, M.; Akanuma, S.; Hosoya, K. Involvement of a novel organic cation transporter in verapamil transport across the inner blood-retinal barrier. Res. 2013, 30, 847-856.
- Kubo, Y.; Shimizu, Y.; Kusagawa, Y.; Akanuma, S.; Hosoya, K. Propranolol transport across the inner blood-retinal barrier: potential involvement of a novel organic cation transporter. Pharm. Sci. 2013, 102, 3332-3342.
- Tega, Y.; Tabata, H.; Kurosawa, T.; Kitamura, A.; Itagaki, F.; Oshitari, T.; Deguchi, Y. Structural requirements for uptake of diphenhydramine analogs into hCMEC/D3 Cells via the proton-coupled organic cation antiporter. Pharm. Sci. 2021, 110, 397-403.
- Lin H, Miller SS. pHi regulation in frog retinal pigment epithelium: two apical membrane mechanisms. Am J Physiol. 1991, 261, C132-142.

Round 2
Reviewer 1 Report
Authors have modified the manuscript according to my observations
Author Response
Response to Reviewer #1
Authors have modified the manuscript according to my observations.
Thank you for the comment. Following the comments from Reviewer #4, we have additionally improved the manuscript by adding Supplementary Figures. It is expected that this revision increases a quality of our manuscript. We are grateful if you also satisfy this revised version.

Reviewer 4 Report
All the concerns in the previous review were appropriately addressed and the revised manuscript now reads well. However, two places show "data not shown", which is not recommended. It would be better to show the data in the supplementary figure.
Author Response
Response to Reviewer #4
All the concerns in the previous review were appropriately addressed and the revised manuscript now reads well. However, two places show "data not shown", which is not recommended. It would be better to show the data in the supplementary figure.
Thank you for your beneficial comment. We are able to agree with your concern, and prepare the supplementary figures to explain that the protein amount between the groups was not significantly different, which is shown as “data not shown” in the previous manuscript. In addition, we re-checked the manuscript and found the mistake in the Discussion section after the revision following your comment. This point was additionally modified in this re-submitted manuscript. We are grateful if you can satisfy these additional revisions.
